# OpenReview forum: "Data Feedback Loops: Model-driven Amplification of Dataset Biases"
_ICLR.cc/2023/Conference — Submitted to ICLR 2023_

### Official Review · Reviewer_8wPf · 2022-10-24

**Confidence:** 3
**Correctness:** 2
**Technical Novelty And Significance:** 3
**Empirical Novelty And Significance:** 3
**Recommendation:** 5

**Clarity, Quality, Novelty And Reproducibility:**

I think this paper has a large room for improvement. For example, the notations could be improved, and many `\vspace{-XX}` could be removed for better readability. During reading this paper, I felt this paper was hard to follow, and I was in trouble understanding the main message of this paper. I strongly recommend revising the paper for better understanding and clarity.

Based on my limited understanding due to the clarity, I think the contribution of this paper is limited because (1) the proposed scenario cannot cover practical settings, such as when $m \rightarrow 0$ and noisy human annotations (2) the experimental results show that the existing algorithms do not suffer from feedback bias. Please correct me if I am wrong.

**Strength And Weaknesses:**

## Strength

- The proposed scenario is practical and could be useful for many practitioners working in the industry. I think the proposed scenario is specifically useful for a recommender system because recommender systems in the real world suffer from the self-feedback problem (or feedback loop).
- The proposed intervention strategy seems to reduce toxicity of a language model.

## Weakness

### Clarity (overused and misused notations for $P$)

This paper is difficult to follow due to inconsistent notations. For example, $P$ was initially defined as a joint probability of data instance $x$ and its corresponding label $y$ (Section 3), but it is often misused as a marginal probability of data instance $x$ only (4.2 Setup and goals), or distribution of dataset $S$ (where $S$ seems the set of the sampled data points). $P$ is also used for the "model-annotated" dataset (P(f)), and $P_\phi$ is used for the expectation. It is very confusing whiling reading the paper, and it should be corrected as the correct notations. For example,

- P is overused. For example, `P: (X, Y) -> R`, `P(x): X -> R`, `P(f): (X, new Y) -> R`, and even $P_\phi$ is used for exepection. I would like to recommend using a different notation for the marginal distribution P(X) or using P(X,Y) for the joint probability. Similarly, P(f) seems very confusing. I would like to recommend using $P^f$ instead.
- I cannot understand why $n_t$ is required to define $P_t^{n_t}$. If P is a distribution, the number of samples is not related to defining the distribution. It is maybe used for defining $S_t$.
- $S_t \sim P_t^{n_t}$ seems wrong. It would be $S_t = [ (x_1, y_1) , \ldots, (x_i, y_i), (x_N, y_N) | (x_i, y_i) \sim P_t^{n_t} ]$ (please replace [ to {. The embedded latex of OpenReview does not show \{ correctly)
- $P_\phi$ seems a not good shorthand for expectation. Please use a different notation for this.

I am still confused about the difference between $Q$ and $P$ in Definition 1. If this paper wants to represent a joint distribution of x and y as Q, please keep using Q.

### Practicalness of the scenario ($m \rightarrow 0$ makes a trivial upper bound for Theorem 1)

As my comment in Strength, the proposed scenario is specifically useful for practical recommender systems. However, in this case, the bound becomes very loose because $k >> m$. Recommender systems notably outperform a randomized item suggestion (e.g., showing much better RMSE for movie rating estimation than a random guess), hence, many industrial applications deploy $m \rightarrow 0$ and $k \rightarrow n_t$ in Theorem 1. This makes the bound becomes trivial (the RHS becomes infinity).

Considering that a machine learning algorithm is usually deployed for automation (as well as performances), it is natural to assume that $m \rightarrow 0$ eventually.

### Experimental results?

I am a bit confused by the experimental results. As far as the reviewer understood, the results may imply that there is no specific amplification by model feedback. For example, in Figure 3 and 4, the blue lines, the empirical one, look very consistent, whereas the orange lines, the theoretical one, are increasing. Figure 5 also shows similar results, but it also provides additional information for the proposed intervention.

If models are not biased by the feedback rounds (but just show better performances), why do we need to consider the proposed scenario? It may show that the proved upper bound is a trivial bound so that it does not align with the empirical results.

### Correctness ($\mathbb E_{f_t}$, and $f_t \sim A (S_t)$)

It could be a minor comment. The quantity of bias amplification (and all its related measures) are defined by the expectation over the function $f_t$ where $f_t$ is drawn from a random distribution. However, first, it is usually impossible to get the correct function space (e.g., how can we get the function space of ResNet-18?). I presume that the expectation notation means that the quantity is measured by the average of different SGD runs

> Here A: (X,Y) -> F refers to a potentially stochastic learning algorithm, which we take to be a neural network trained on the cross entropy loss with SGD

In this case, I recommend avoiding using $\mathbb E_{f_t}$, and $f_t \sim A (S_t)$, because not all learning algorithms are stochastic unless the stochasticity of the algorithm is crucial to the theorem proof. For example, KNN is not stochastic (unless we consider a tie-breaking). Instead, it could be better to treat $f$ can be obtained in a deterministic way, and clarify that the experiments report the average of XX different SGD runs to reduce the randomness.

### Practicalness of the scenario ("clean human annotation" assumption)

It could be a minor concern, but I am concerned about the basic assumption that human annotations are clean ("clean human-annotated samples are available on the internet" Section 3). In practice, (especially in recommender systems) human annotations and human activities are very noisy. It makes the existing recommender systems use implicit feedback instead of explicit feedback [R1]. Because this paper focuses on a theoretical foundation of the data feedback problem, it could be treated as a "strong assumption" that does not match real-world situations. However, I think it should be carefully discussed in this paper, even if this paper does not have to show an additional bound for the noisy human annotation scenario.

[R1] Hu, Yifan, Yehuda Koren, and Chris Volinsky. "Collaborative filtering for implicit feedback datasets." 2008 Eighth IEEE international conference on data mining. Ieee, 2008.

**Summary Of The Paper:**

This paper proposes a new scenario called "data feedback". This scenario assumes a multi-round annotation process, where at each round, the new annotations are annotated by both humans and models. This paper provides a quantity of how the new annotated datasets and a trained model by the dataset will be biased to the model annotations, named "consistently calibration". This paper also proposes the "consistently calibrated learning algorithm" defined by Theorem 1. The experiments are conducted in image classification, visual role labeling and language models.

**Summary Of The Review:**

I think this paper needs a heavy revision for improvement. Especially, I think this paper should be polished in terms of clarity and presentation. If the reviewer understood the paper correctly, this paper has a limited novelty. Overall, I recommend "reject".

---

Post-rebuttal comment: As my last comment, the message of the proposed theorem and experiments are still confusing to me. I would like to suggest polishing the main message in the next revision. I think this paper could be valuable in some sense, but I think this paper has a large gap for improvement in terms of clarity, presentation, and writing. I encourage the authors to revise the paper to make the overall message clearer.

---

> ### Author Response · Authors · 2022-11-16
> **Author response to reviewer 8wPf (1/3)**
>
> We thank the reviewer for their detailed feedback, have provided an updated manuscript, and address their specific concerns below.
>
> > The proposed scenario is practical and could be useful for many practitioners working in the industry. I think the proposed scenario is specifically useful for a recommender system because recommender systems in the real world suffer from the self-feedback problem (or feedback loop).
>
> While the data feedback setting does have connections to recommender systems (detailed in Appendix A, page 14), this work addresses problems that are quite distinct from the recommender system setting. Our work focuses on the setting where the annotations for an example come from either model predictions or a human, which is a reasonable model in settings like translation, image tagging, text-to-image generation, etc. In recommender systems, the rating is always produced by a human, but the selection of which items are rated is a function of the recommendation model. This is an important difference which we will elaborate on further below.
>
> > The proposed intervention strategy seems to reduce toxicity of a language model.
>
> Our proposed intervention reduced the repetition bias of the language model, but it actually increased its toxicity. This is since the original model’s toxicity level was lower than the dataset level, thus lowering the consistently calibration error by increasing its generation toxicity. It is worth noting that our bounds and characterizations hold even for counterintuitive situations such as these.
>
> > This paper is difficult to follow due to inconsistent notations.
>
> We agree that the notation could have been cleaner, and so we have updated the main text of the manuscript (updates in blue) with the reviewer’s suggestions. We thank the reviewer for giving detailed feedback to help improve the readability of the paper. We opted to keep the expectation shorthand $\mathsf P \phi$ for the proofs in the appendix to keep them concise - the shorthand is standard operator notation [see https://people.eecs.berkeley.edu/~jordan/sail/readings/kosorok.pdf, page 10 bottom], and it should help with readability for the proofs (we also redefine the shorthand in the appendix for the reader).
>
> > I am still confused about the difference between $\mathsf Q$ and $\mathsf P$ in Definition 1. If this paper wants to represent a joint distribution of x and y as Q, please keep using Q.
>
> $\mathsf Q$ is not the joint distribution of x and y over which we study data feedback. This joint is represented by $\mathsf P(x, y)$ and we have updated the text accordingly. $\mathsf Q$ is part of the condition in Definition 1. Def 1 states the condition must be met for *all* distributions $\mathsf Q(x, y)$ that match the marginal $\mathsf P(x)$. Data feedback is performed starting from $\mathsf P(x, y)$, and def 1 is a property that holds for all hypothetical distributions $\mathsf Q(x, y)$, which is why we used a separate character for it.

---

> ### Author Response · Authors · 2022-11-16
> **Author response to reviewer 8wPf (2/3)**
>
> > As my comment in Strength, the proposed scenario is specifically useful for practical recommender systems. However, in this case, the bound becomes very loose because $k > m$. Recommender systems notably outperform a randomized item suggestion (e.g., showing much better RMSE for movie rating estimation than a random guess), hence, many industrial applications deploy $m \rightarrow 0$ and $k \rightarrow n_t$ in Theorem 1. This makes the bound become trivial (the RHS becomes infinity). Considering that a machine learning algorithm is usually deployed for automation (as well as performances), it is natural to assume that eventually.
>
> We would like to clarify that the takeaways from our work cannot be immediately ported into the recommender systems setting. The big difference is that in data feedback, annotations are collected from both humans and model predictions, while the distribution of examples for which the annotations are collected remains fixed. In recommender systems, the annotation/rating is always produced by a human, and the distribution of items for which the rating is collected is a function of the recommendation model. It is *not* true that $m \rightarrow 0$ recovers the recommender systems setting; this is a setting where the distribution over examples is fixed and the annotations come only from model predictions. In recommender systems, the distribution over examples itself is changing as a function of the model, which violates the fixed covariate assumption of data feedback (first paragraph of page 4). In addition, annotations are only collected from humans, not a mix of humans and model predictions. We would appreciate it if the reviewer evaluated this work based on the different real-world scenarios under which data feedback is a plausible model (machine translation, image tagging, story writing, etc.) instead of for recommender systems.
>
> Finally, yes, even in these realistic situations, $m \rightarrow 0$ may happen eventually. In these cases, our bound does become vacuous, but this is also unavoidable without more assumptions. In particular, imagine a binary classification problem with a slight label imbalance. Say our learning algorithm is such that it predicts the majority class accurately but mislabels 10% of the minority class as majority. Even though this model satisfies consistent calibration, if $m = 0$ then over time the prevalence of the majority class will keep increasing until the whole dataset is basically majority class and the model only predicts the majority class.
>
> > I am a bit confused by the experimental results. As far as the reviewer understood, the results may imply that there is no specific amplification by model feedback. For example, in Figure 3 and 4, the blue lines, the empirical one, look very consistent, whereas the orange lines, the theoretical one, are increasing. Figure 5 also shows similar results, but it also provides additional information for the proposed intervention.
>
> Our results show that amplification is dependent on the initial bias of the model and whether the model behaves like a sampler (i.e. has a consistent calibration guarantee). In Figure 3, the lines look consistent since the models have a consistent calibration guarantee and are thus stable. But in Figure 4, the initial bias is quite large and is further quickly amplified +5-7% during feedback, as stated in the figure caption. A read of the y-axis (as mentioned in the caption) is necessary to contextualize the results. In Figure 5 right, there is severe repetition bias amplification for the beam search model (blue). We would appreciate any clarifications the reviewer may have on where amplification does not exist in these figures.
>
> > If models are not biased by the feedback rounds (but just show better performances), why do we need to consider the proposed scenario? It may show that the proved upper bound is a trivial bound so that it does not align with the empirical results.
>
> There is one case where the upper bound does not align with empirical results - in Figure 5 right, the beam search model exceeds the bound. But this is discussed in the main text - Theorem 1 isn’t applicable in this case as beam search models do not have a consistent calibration guarantee. In the rest of the figures, Theorem 1 is indeed a useful upper bound, as demonstrated empirically. In particular, in Figure 3, we go on to stress test our upper bound with a certain worst-case experimental condition (shown in gray), and we find the upper bound still holds and qualitatively captures the amplification behavior well. We would appreciate any clarification from the reviewer as to how the upper bound is trivial or does not align with empirical results.

---

> ### Author Response · Authors · 2022-11-16
> **Author response to reviewer 8wPf (3/3)**
>
> > I presume that the expectation notation means that the quantity is measured by the average of different SGD runs.
>
> Yes, this is correct, and we have updated the main text with this clarification.
>
> > In this case, I recommend avoiding using $\mathbb E_{f_t}$, and $f_t \sim \mathcal A(\mathcal S)$, because not all learning algorithms are stochastic unless the stochasticity of the algorithm is crucial to the theorem proof.
>
> The stochasticity of the algorithm is not crucial to the proof - we explicitly say that $\mathcal A$ is *potentially* stochastic and use the $f_t \sim \mathcal A(\mathcal S)$ formalism to account for cases where it is. The expectation $\mathbb E_{f_t}$ denotes an expectation not only over randomness in model training but also for sampling new examples throughout data feedback, as is explained at the bottom of page 3. The expectation is therefore needed whether or not the learning algorithm is stochastic. No part of the proofs or definitions rely on the stochasticity (or determinism) of the training algorithm.
>
> > I am concerned about the basic assumption that human annotations are clean ("clean human-annotated samples are available on the internet" Section 3). In practice, (especially in recommender systems) human annotations and human activities are very noisy. It makes the existing recommender systems use implicit feedback instead of explicit feedback [R1]. Because this paper focuses on a theoretical foundation of the data feedback problem, it could be treated as a "strong assumption" that does not match real-world situations. However, I think it should be carefully discussed in this paper, even if this paper does not have to show an additional bound for the noisy human annotation scenario.
>
> The “clean” in this sentence simply means that the examples are annotated only by humans and not models (and thus have not been impacted by any data feedback effects). We understand this may be confusing, so we have removed the word in the main text. None of the assumptions, theoretical, or experimental parts of the paper rely on having clean/non-noisy annotations, and so noisy examples are not a problem. In fact, the CIFAR-5m dataset we use in Section 5.1 is a noisy dataset since the examples are generated & labeled synthetically. None of the theoretical bits change either - the main assumption is Def 1 (consistent calibration), which can definitely hold when the dataset contains noisy examples as well.
>
> > Based on my limited understanding due to the clarity, I think the contribution of this paper is limited because (1) the proposed scenario cannot cover practical settings, such as when $m \rightarrow 0$ and noisy human annotations (2) the experimental results show that the existing algorithms do not suffer from feedback bias. Please correct me if I am wrong.
>
> We believe these criticisms stem from some misunderstandings of our work that we clarified above:
>
> 1. $m \rightarrow 0$ is currently only a practical setting for recommender systems, which are already not a fit for this framework due to the differences in setup outlined above. In addition, noisy human annotations pose no challenge for our framework - none of our theoretical assumptions require this, and indeed we have experiments with noisy data.
>
> 2. The experimental results show that existing algorithms can and *do* suffer from feedback bias. Look to gender bias (Figure 4) and beam search repetition bias (Figure 5 right) for the best examples. Our work gives the tools necessary to understand under which experimental settings bias amplification is a problem, to characterize how much it can amplify when models are relatively stable (Theorem 1, Figure 3), and to stabilize feedback systems which may be unstable (Figure 5).

---

> > ### Comment · Reviewer_8wPf · 2022-11-21
> > **Response**
> >
> > Thanks for your comments and sorry for the late delay.
> >
> > First, thanks for pointing out that the recommender system is not the only case for this problem, and it could be slightly different from the target problem (e.g., "annotations" for recommender systems are made by humans). I thought about the case where a recommender system iteratively shows new recommendation items based on their previous recommendations. It is a well-known problem in recommender systems, but after reading the response, I agree with the authors; it is not directly linked to the proposed formulations.
> >
> > Second, I also agree that noisy labels by human annotators can be out-of-scope in this paper. I think it is still an important problem, and the proposed formulations could be easily extendable to this scenario, but I agree that it is out-of-scope of this paper.
> >
> > Third, I appreciate the authors clarifying my misunderstandings. I still feel the paper is not easy to follow, and the message is still unclear, but, I found that other reviewers mentioned that the paper is easy-to-follow or readable (Reviewer vb9Z also pointed out notations). My lack of expertise in this area could be a reason. However, I am still confused about the results. For example:
> >
> > > Our results show that amplification is dependent on the initial bias of the model and whether the model behaves like a sampler (i.e. has a consistent calibration guarantee). In Figure 3, the lines look consistent since the models have a consistent calibration guarantee and are thus stable. But in Figure 4, the initial bias is quite large and is further quickly amplified +5-7% during feedback, as stated in the figure caption. A read of the y-axis (as mentioned in the caption) is necessary to contextualize the results. In Figure 5 right, there is severe repetition bias amplification for the beam search model (blue). We would appreciate any clarifications the reviewer may have on where amplification does not exist in these figures.
> >
> > I am still confused. The blue lines (the empirical results) in Figure 3, 4, 5 have a large gap (with a different asymptotic complexity) between the orange lines (the theoretical upper bound). It may show that the theoretical upper bound could be a trivial bound, i.e., the upper bound can be a loose bound. I argue that the trends between blue and orange lines seem to have different asymptotic complexity: Orange lines keep increasing, but blue lines are quickly saturated (even in Figure 4 and 5).
> >
> > Fourth, after reading the revised paper and the response, I partially agree that $m \rightarrow 0$ could be less practical than I expected the first time. I don't think $m \rightarrow 0$ is only a practical setting for recommender systems, but it is also a practical scenario for other practical systems based on ML automation. Especially because human annotations are much more expensive than machine annotations, in practical applications, $\frac{m+k}{m}$ could become a trivial upper bound.
> >
> > The response addressed my questions, and it helped my understanding of the paper. I agree that some concerns in my initial review could be out-of-scope or less important than I expected (e.g., noisy human labels and $m \rightarrow 0$ case). I still have concerns about clarity and quality, but I would like to mention that my limited expertise could be the reason (as my 3rd comment).
> >
> > I revised my score from 3 to 5 because my concerns are based on my misunderstanding. I did not revise my score to 5 because I still think the experimental results and the theoretical results are not easily connected.

---

> > > ### Author Response · Authors · 2022-11-29
> > > **Updated author response (1/2)**
> > >
> > > We thank the reviewer for taking the time to engage with our response and review the updated manuscript. We are happy to continue the dialogue and appreciate the reviewer’s time.
> > >
> > > > I am still confused. The blue lines (the empirical results) in Figure 3, 4, 5 have a large gap (with a different asymptotic complexity) between the orange lines (the theoretical upper bound). It may show that the theoretical upper bound could be a trivial bound, i.e., the upper bound can be a loose bound. I argue that the trends between blue and orange lines seem to have different asymptotic complexity: Orange lines keep increasing, but blue lines are quickly saturated (even in Figure 4 and 5).
> > >
> > > We think the behaviors seen in figures 3, 4, 5 show that the upper bound is surprisingly tight. Please allow us to elaborate further for each graph mentioned:
> > > - In Figure 3, the orange line (our bound) shows that if calibration errors do not decrease with dataset size, the bias may amplify rapidly. The *gray lines* show that this bound is quite tight under the assumptions of the bound. The blue line is lower, but this is due to the fact that the model becomes *better calibrated* due to a larger sample size. As we note in Appendix C.1, the gray line shows that we cannot make the bound tighter without making stronger assumptions on model performance -- e.g. assuming that calibration improves as some function of the dataset size. We note that this may be possible using techniques such as scaling laws, but we considered this out of scope for this work.
> > > - In Figure 4, the orange bound is tight up to the point where it exceeds 100%, at which point the bound predicts that the model may become fully degenerate and predict 100% female. The blue line shows that this is what happens. We agree that we could have tightened the bias amplification bound for bounded bias metrics by considering min(our_upper_bound, max_value_of_bias), but did not do so as it added complexity to the bound without changing the qualitative conclusions
> > > - For Figure 5 - as we state in the text, figure 5 is *not* intended to show that the upper bound matches the observed bias. Instead, the experiment in figure 5 analyzes the behavior of data feedback when the assumptions to Lemma 4.1 / Definition 1 are violated. In this case, we expect the bound to not hold, and that's what we see in figure 5. The fact that the orange line is below the blue should be viewed as a positive result that shows the need for consistent calibration.
> > >
> > > We believe that in these figures 3-5, the qualitative behaviors of data feedback clearly agree with the predictions from the bounds. The reviewer's request to have the tight, quantitative agreement where the orange and blue lines always agree is beyond the norm in the vast majority of the field -- consider, for example, Rademacher complexity or H-delta-H bounds, both of which are conceptually useful but very loose for most settings. Previous theoretical results in settings similar to data feedback (such as performative prediction) have only been studied in convex settings or with Gaussian data. Being able to derive non-trivial predictions in more complex settings is a strength of our work, in our view.

---

> > > ### Author Response · Authors · 2022-11-29
> > > **Updated author response (2/2)**
> > >
> > > > Fourth, after reading the revised paper and the response, I partially agree that $m \rightarrow 0$ could be less practical than I expected the first time. I don't think $m \rightarrow 0$ is only a practical setting for recommender systems, but it is also a practical scenario for other practical systems based on ML automation. Especially because human annotations are much more expensive than machine annotations, in practical applications, $\frac{m+k}{m}$ could become a trivial upper bound.
> > >
> > > Yes, we agree with the reviewer that the fraction of model-annotated data can increase quite rapidly. The takeaway here isn’t that $\frac{m+k}{m}$ is a trivial upper bound, but that our work provides the tools to help navigate the tradeoffs that arise. As the reviewer suggests, consider the case where the fraction of model-labeled data is unbounded. In this case, the model developer can choose to not update the model, make updates more infrequent, or continually update the model on self-annotated data:
> > > - Not updating the model mitigates bias amplification, but the overall utility (e.g. accuracy) of the model may be lower than if otherwise trained on all available data, as we discuss in Section 5.1 (page 7 para 2).
> > > - Making updates more infrequent can also mitigate amplification - the Theorem 1 time-dependent bound (middle quantity). The intuition here is that though the same total amount of data may be used for training, making updates slower reduces how frequently errors can compound, which Theorem 1 predicts.
> > > - Continually updating a model on self-annotated data is inherently unstable (the majority/minority example provided in our last reply). That $\frac{m+k}{m}$ becomes trivial here is reasonable and expected. To help clarify this, we will add a lower bound in the appendix showing that there are fundamental tradeoffs between updating a model and ensuring stability.
> > >
> > > In addition to these choices, model deployers may choose to deploy watermarking or advanced data filtering methods to detect which samples are model-annotated or human-annotated. In this case, the quantities of data fed into the model ($m$ and $k$) may differ drastically from the data quantities that exist in the wild.
> > > Lastly, we note that the time-dependent bound we present grows much slower than the time-independent $\frac{m+k}{m}$. One way to see this is in Figure 3, where even when the vast majority (80%) of all samples are model-labeled, the upper bound is non-trivial - only an excess of 2-4% over 90 feedback rounds.
> > >
> > > To help clarify these takeaways, we will add a section in the appendix mirroring this discussion.
> > >
> > > > I did not revise my score to 5 because I still think the experimental results and the theoretical results are not easily connected.
> > >
> > > Please let us know if our response above helped to clarify this concern, as we do think a strength of our work is connecting the theoretical framework and predictions to the experimental results.

---

> > > > ### Comment · Reviewer_8wPf · 2022-12-05
> > > > **Response**
> > > >
> > > > Thanks for your additional comments. However, I am still skeptical about revising my score to acceptance due to the following reasons:
> > > >
> > > > - The experimental results are still confusing to me.
> > > >     - "The blue line is lower, but this is due to the fact that the model becomes better calibrated due to a larger sample size". It could be, but is there any guarantee that the blue line is actually calibrated better? I couldn't find any connection between the blue line and better calibration after re-reading the paper. It is not directly revealed in the current text. Yes, it could be, but I think that it needs more verification. For example, showing the number of samples vs. calibration and the corresponding bias measures.
> > > > - I partially agree that the tight bound is not always achievable. Rademacher complexity and H-delta-H bounds are loose, and the variational bound is also loose. Even many traditional regret bounds are loose. However, I am not just arguing that the theorem is not good because the bound is loose; my concern is that the intuition by the theorem could be misleading because the bound is not directly linked to the realistic scenarios as this paper aims to.
> > > >     - At least from my point of view, the blue lines in Figure 3 and 4 look stable. While the orange upper bound is still increasing. It would mean that, in practice, we do not have to care about bias amplification if we have a large size of the dataset or if the initial bias measure is too large. I think the proposed upper bound could be useful, but the experimental results (as well as the practical scenario) and the upper bound are not directly linked for me.
> > > > - Finally, I feel that the message of the current revised paper and the author's comment are not directly aligned. I know that it is not possible to revise the paper now, but it seems that the paper needs a heavy revision to re-align the messages, re-presenting the overall experimental results and corresponding interpretations and messages. In my opinion, this paper should improve its clarity.
> > > >
> > > > Overall, the message of the proposed theorem and experiments are still confusing to me. I would like to suggest polishing the main message in the next revision. In particular, I'd like to suggest revising Section 5 to make each subsection based on the message rather than different modalities (image classification, ...).
> > > >
> > > > I think this paper could be valuable in some sense, but I think this paper has a large gap for improvement in terms of clarity, presentation, and writing. I encourage the authors to revise the paper to make the overall message clearer.

---

### Official Review · Reviewer_Dg6m · 2022-10-25

**Confidence:** 4
**Correctness:** 3
**Technical Novelty And Significance:** 2
**Empirical Novelty And Significance:** 4
**Recommendation:** 6

**Clarity, Quality, Novelty And Reproducibility:**

The clarity is good and the writing is easy to follow.
The novelty is good, mainly the formulation of the problem. The theoretical analysis has a lot of room for improvement.

**Strength And Weaknesses:**

Strength: The targeted problem is very interesting and perhaps has been overlooked for practical AI applications. For this under-studied problem (to the best of my knowledge), the formulation/definition and the numerical experiments setup are very helpful and may inspire more investigations along this line.

Weakness:
1. Though the motivation of the problem is new to me, the technical problem itself is not completely new in literature.
Some related works are missing, e.g., self-distillation, self-training, classification calibration, etc.
I believe a lot of existing results can be cross-referenced.


2. Though the formulation is clear to me, the analysis does not provide enough insight in my opinion.
The consistent calibratin (df 1) and distinguishability definition seemed artificial and does not go deep enough to properties of the data or training algorithm themself. For example, in the simplest case where t=1 (similar to the typical teacher-student learning setting), when will the teacher be bad for the student? In comparison, derivation from t=1 to infinity is more straightforward.

Some questions:
The authors recommend the classifier to overfit the training labels, e.g., running more iterations to get the training error to be smaller. However, predictions for unseen data x to be labeled may not be well-calibrated, due to the tradeoff between bias and variance. Do we want the Bayes optimal classifier in this case? When classes are not separable, the Bayes classifier does not interpolate.
How would typical classification calibration method work in the case of bias amplification?



**Summary Of The Paper:**

This paper investigates a very interesting problem, how artificial labels affect later model performance if the predicted labels are recorded as training data. The authors analyzed the stability and gave sufficient conditions to control the bias amplification. Numerical experiments are conducted to corrobarate the theoretical analysis.

**Summary Of The Review:**

Overall, I think this work is inspiring and original. The bias amplification due to model-labeling is an important problem. This paper analyzed under what conditions the amplification effect can be controlled and conducted various experiments for verification. Connections to related work and the theorectical analysis have a lot of room for improvement.

---

> ### Author Response · Authors · 2022-11-16
> **Author response to reviewer Dg6m (1/2)**
>
> We thank the reviewer for their feedback, have provided an updated manuscript, and address their specific concerns below.
>
> > Some related works are missing, e.g., self-distillation, self-training, classification calibration, etc. I believe a lot of existing results can be cross-referenced.
>
> We have an additional related work discussion in Appendix A, which includes semi-supervised learning/self-training, and we have also added a section on classification calibration error. Please let us know if there are any other papers which may be relevant and we would be happy to update the paper.
>
> > Though the formulation is clear to me, the analysis does not provide enough insight in my opinion. The consistent calibratin (df 1) and distinguishability definition seemed artificial and does not go deep enough to properties of the data or training algorithm themself.
>
> We would appreciate it if the reviewer could elaborate on what they mean by “does not go deep enough to properties of the data or training algorithm”. We chose the consistent calibration definition as a high-level abstraction that can characterize many settings, including some previously studied in the literature. For example, the same framework allows us to study label bias in image classification as well as repetition bias in prompted language generation. The benefit of this approach is that it allows a lot of room for further experimental design, which we see as a strength. In particular, we were able to successfully leverage the Distributional Generalization (DG) intuition from image classification over into language generation, a very different domain, for our intervention. Incorporating specifics of the training data or algorithm may be asking a lot out of theory, but we do think a more thorough understanding of how different experimental choices affect consistent calibration error would be great future work.
>
> > For example, in the simplest case where t=1 (similar to the typical teacher-student learning setting), when will the teacher be bad for the student? In comparison, derivation from t=1 to infinity is more straightforward.
>
> It depends on what the reviewer means by “will the teacher be bad for the student?”. Our analysis suggests that, given a moderate dataset size, bias will increase when teacher examples are added into the student training dataset. (The qualifier of a moderate dataset size is needed since consistent calibration errors will decrease as the dataset size increases. If the initial dataset size is too small, this this reduction in consistent calibration error can outweigh the increase in bias from incorporating teacher examples, as demonstrated experimentally in Figure 14 right). On the other hand, overall utility metrics such as accuracy may improve over time during feedback, which is consistent with the self-training/semi-supervised learning story (Figure 6). Thus many feedback settings may present such a utility-bias tradeoff when including teacher examples, and our work focuses on characterizing the bias amplification side of this.

---

> ### Author Response · Authors · 2022-11-16
> **Author response to reviewer Dg6m (2/2)**
>
> > Some questions: The authors recommend the classifier to overfit the training labels, e.g., running more iterations to get the training error to be smaller. However, predictions for unseen data x to be labeled may not be well-calibrated, due to the tradeoff between bias and variance. Do we want the Bayes optimal classifier in this case? When classes are not separable, the Bayes classifier does not interpolate. How would typical classification calibration method work in the case of bias amplification?
>
> These are great questions. We would like to first clarify that our notion of calibration (consistent calibration) is different from the traditional notion of model calibration. While traditional calibration measures the difference between a model’s predictive probability and its accuracy, consistent calibration measures the difference between a bias statistic computed over model outputs and the same statistic computed on the training distribution (we have added a clarification for this in Appendix A, page 14). The two notions are unrelated; in particular, traditionally calibrating a classifier does not change its consistent calibration error since it does not change model labels.
>
> We agree that overfitting presents downsides, and we have included a more thorough analysis of the utility of our overfitting intervention compared to the other two language models in Appendix C.2, page 18. In particular, we find that the coherence of model generations, measured as the similarity between prompts and model completions, is significantly decreased for the overfit model - 0.26 vs 0.35. This setting presents another way in which the utility-bias tradeoff manifests, as the intervention has lower repetition bias but also less coherent completions. However, there is not always a tradeoff, as well-tuned interpolating classifiers have lower bias (DG gives consistent calibration) and typically have higher test accuracy.
>
> And yes, we do not always want the Bayes-optimal classifier. By training for a long time, we can produce an interpolating classifier (given a large enough model), and DG says that interpolating classifiers actually behave like samplers. But if the model is trained for fewer iterations, it behaves more like a Bayes-optimal classifier; in the extreme case, the model only sees each data sample once, and in this underfit limit, the model does not interpolate when classes are not separable, as the reviewer states. The tradeoff comes down to the setting - in image classification, we can train interpolating classifiers (which give feedback stability) without any generalization penalty, while for beam search language generation, interpolation comes at the price of coherent generations.
>
> > Connections to related work and the theoretical analysis have a lot of room for improvement.
>
> Please let us know of any additional related works and along which axes the theoretical analysis may be improved. In this work, our focus started with different real-world feedback settings, and we introduced just enough theoretical abstraction (e.g. consistent calibration) to derive useful upper bounds and help characterize our experiments.

---

> > ### Comment · Reviewer_Dg6m · 2022-11-27
> > **Response**
> >
> > Thanks for providing the detailed clarifications and sorry for the late reply. My initial concerns on related work and classification calibration have been addressed. However, I still think the theoretical analysis part is too general and the provided insights are relatively weak. I will remain my initial score.

---

> > > ### Author Response · Authors · 2022-11-29
> > > **Updated author response**
> > >
> > > We thank the reviewer for taking the time to read and engage with our response.
> > >
> > > > I still think the theoretical analysis part is too general and the provided insights are relatively weak.
> > >
> > > We view the strength of this work as connecting the theoretical model and predictions to the experimental results. To get a better understanding, could the reviewer elaborate on which insights were found to be weak?

---

> > > > ### Comment · Reviewer_Dg6m · 2022-12-08
> > > > **Response**
> > > >
> > > > Let me clarify my perspective.
> > > > I think the problem this paper studies is very interesting and perhaps under-explored in literature. After the formulation, I am intrigued to see under what circumstances will the feedback be well controlled. However, the theoretical analysis is not comprehensive enough in my opinion. To be more specific, after definition 1, theorem 1 is very natural, and so is lemma 4.1 after defining ($\delta,\mathcal{A},P(x), n$)-distinguishable.
> > > > To me, these results shift one question to another, finding sufficient conditions, but do not answer the question in a more intuitive and verifiable way, e.g., what data/algorithm, under what scenario will the distinguishable condition be satisfied is still not clear.
> > > > Of course, I am not denying the contribution of linking bias amplification to consistent calibration, and consistent calibration to DG.
> > > > I voted for acceptance for this work but due to the limitation of the current analysis, I am very skeptical about raising my score.

---

### Official Review · Reviewer_vb9Z · 2022-10-25

**Confidence:** 4
**Correctness:** 3
**Technical Novelty And Significance:** 3
**Empirical Novelty And Significance:** 3
**Recommendation:** 5

**Clarity, Quality, Novelty And Reproducibility:**

While not the clearest paper, it is readable and the analysis novel. Should be reproducible with the details provided.

**Strength And Weaknesses:**

Strengths
- Interesting problem on a likely an increasingly common scenario going forward.
- Tested on biases in both computer vision (label bias, gender bias) and language (toxicity and repetition)
- Takeaways in figures is a nice addition

Weaknesses
- paper is quite short in content-- exposition on experiments could be moved to appendix and more results and analysis presented in the main paper.
- While the bound is helpful and this experiment seems like it would yield an obvious result, nonetheless it would be nice to see an ablation of how this problem is exacerbated by different data fraction sizes.
- While I appreciate that Distributional Generalisation does catastrophically fail as Bayes-optimial classifier would in the scenario-- it doesn't mitigate against bias initially present; and seems like this shouldn't be presented as a solution. Would be interested to hear some exposition around this.

Minor comments
- Colours chosen in figures make it hard to tell what is what at time.
- Notation is somewhat confusing to follow in section 4 (e.g. use of shorthand and consistency). I will add more detailed feedback here, but believe this could be easily fixed.

**Summary Of The Paper:**

This paper analyses the effect of dataset bias amplification when training a model it's own outputs (with it's biases at that time), which seems like likely scenario given direction of machine generated content. The paper provides bound in a simplistic scenario of data feedback loop and analyses this in a number of modalities and in the context of a number of biases.

**Summary Of The Review:**

Good analysis, if somewhat short, on an interesting and increasingly prevalent problem. While not a great paper yet, I see no reason to reject if the weaknesses mentioned are addressed.

---

> ### Author Response · Authors · 2022-11-16
> **Author response to reviewer vb9Z**
>
> We thank the reviewer for their feedback, have provided an updated manuscript, and address their specific concerns below.
>
> > paper is quite short in content-- exposition on experiments could be moved to appendix and more results and analysis presented in the main paper.
>
> Please let us know which results and/or analysis would benefit from more exposition and we would be happy to include more or to re-organize.
>
> > While the bound is helpful and this experiment seems like it would yield an obvious result, nonetheless it would be nice to see an ablation of how this problem is exacerbated by different data fraction sizes.
>
> Thank you for the suggestion! We added experiments with 20% and 5% of new samples being model-labeled, both for image classification (Figure 15) and for language generation (Figures 21 & 22). Appendix F also includes ablations for different choices such as: model architecture, training data distribution, initial dataset size, and more.
>
> > While I appreciate that Distributional Generalisation does catastrophically fail as Bayes-optimal classifier would in the scenario-- it doesn't mitigate against bias initially present; and seems like this shouldn't be presented as a solution. Would be interested to hear some exposition around this.
>
> We would like to clarify that Distributional Generalization (DG) does not fail as a Bayes-optimal classifier would. In fact, DG states that interpolating classifiers behave like samplers - in particular, that they are consistently calibrated, which gives the models feedback stability. If a model was perfectly DG, then it would have a consistent calibration error of 0, and so there would be no bias to amplify.
>
> In general, there are 2 independent axes for stability: 1) how small the initial bias is ($\delta_{n_0}$), and 2) whether the model is consistently calibrated*. If the model is not consistently calibrated (e.g. does not behave like a sampler), then Theorem 1 does not apply and may be inaccurate - for example, the beam search model greatly exceeds its repetition bias bound in Figure 5 (right). But even if the model is consistently calibrated, if the initial bias is large, bias may also amplify quickly (though within the limits of the bound) - as in the gender bias experiments in Figure 4.
>
> The presented overfitting intervention does greatly reduce the initial repetition bias present with beam search models (blue to red in Figure 5 right). We would appreciate any clarification on the statement “it doesn't mitigate against bias initially present; and seems like this shouldn't be presented as a solution”.
>
> *Any training procedure is consistently calibrated with a large enough error. What we mean here is that the gap between the measured initial bias and the “true” consistent calibration error is not too large.
>
> > Colours chosen in figures make it hard to tell what is what at time.
>
> Thanks for pointing this out. We have updated the colors for Figure 5. Please let us know if you have any additional feedback.
>
> > Notation is somewhat confusing to follow in section 4 (e.g. use of shorthand and consistency). I will add more detailed feedback here, but believe this could be easily fixed.
>
> We have adjusted the notation in the main text to be more consistent and have eliminated some shorthands. If the reviewer has any additional feedback, we would be happy to incorporate that as well.

---

### Official Review · Reviewer_GiPd · 2022-10-26

**Confidence:** 3
**Correctness:** 3
**Technical Novelty And Significance:** 3
**Empirical Novelty And Significance:** 3
**Recommendation:** 5

**Clarity, Quality, Novelty And Reproducibility:**

- Clarity: The paper is overall clearly written and easy to follow

- Quality and Novelty: The paper studies a novel problem setting with decent theoretical framework/analysis. Empirical results are less strong, which mainly focuses on validating the theoretical findings.

- Reproducibility: The authors provide implementation details in the paper as well as their code to reproduce the experimental results

**Strength And Weaknesses:**

Strengths:
- I believe the general idea of data feedback loop is a pretty timely topic to study, where bias amplification is one important angle to look into. The paper does a good job in formalizing a clean framework for this to allow further theoretical analysis.
- The paper organization is easy to read, and I appreciate the authors include an illustrative example to motivate the analysis.

Weaknesses:
- While the theoretical framework is clean and the results make sense. The main takeaways from the theoretical analysis appear to be less surprising. Intuitively, it is quite straightforward that one can mitigate bias amplification by either (1) adding more human-annotated examples or (2) calibrating model’s prediction w.r.t. the original data training distribution. Could the author provide more insights and/or contributions here?
- I appreciate that the proposed framework enables a way to measure bias amplification in the data feedback loop. However, I think intervention (how to avoid the amplication) is one key that readers would be interested in. However, it is only touched on in the very last section of the paper, and not discussed deeply. For example, by simply overfitting the model, does it also affect the model’s generalization ability, i.e., accuracy?
- Similar to the above, if sampling-based models are less prone to amplify bias, how does these models compare to argmax models on performance metrics? Is there a trade-off between the two metrics? This is not currently discussed, validated in the experiments.
- Sec 5.2 confuses me a bit about the usefulness of the proposed theoretical framework. Does it mean that the proposed bound cannot provide useful insights on real-world datasets where the calibration error can frequently be high?
- In Sec 5.3, isn’t the lower toxicity level the better? If so, why should we care about mitigating the amplification in the cost of resulting in a more toxic model?

Minor Questions:
- Experiments are shown in the settings with 80% and 50% of new samples are model-labeled. How about the settings where <50% are model-labeled?


**Summary Of The Paper:**

The paper formalizes and studies the problem of “data feedback loop” where model-labeled data is added into the training dataset iteratively. Specifically, the paper focuses on investigating whether dataset biases would be amplified with the data feedback loop. A theoretical framework was proposed to study the problem, which characterizes the conditions where the bias amplification is likely to occur. The theoretical insights are justified by both synthetic and real-world datasets. Finally, a method is proposed to help mitigate the bias amplification.

**Summary Of The Review:**

Overall, I like the problem setting and how the authors approach the problem with a clean framework. However, I hope to see more takeaways/insights to be provided from the theoretical analysis and more experimental results outside of only validating the theoretical findings. Specifically, how one can leverage the findings to design better methods to solve real-world bias amplification problem.

---

> ### Author Response · Authors · 2022-11-16
> **Author response to reviewer GiPd (1/2)**
>
> We thank the reviewer for their detailed feedback, have provided an updated manuscript, and address their specific concerns below.
>
> > The main takeaways from the theoretical analysis appear to be less surprising. Intuitively, it is quite straightforward that one can mitigate bias amplification by either (1) adding more human-annotated examples or (2) calibrating model’s prediction w.r.t. the original data training distribution. Could the author provide more insights and/or contributions here?
>
> We would like to first clarify that our notion of calibration (consistent calibration) is different from the traditional notion of model calibration, and this presents interesting insights and opportunities for algorithm design. While traditional calibration measures the difference between a model’s predictive probability and its accuracy, consistent calibration measures the difference between a bias statistic computed over model outputs and the same statistic computed on the training distribution (we have added a clarification for this in Appendix A, page 14). The two notions are unrelated; in particular, traditionally calibrating a classifier does not change its consistent calibration error since it does not change model labels.
>
> Given the consistent calibration definition, one could think of training or post-training adjustment strategies to lower the consistent calibration error of the model. However, this requires knowing the bias metric ahead of time, before any feedback effects are experienced. In contrast, our overfitting intervention is designed to broadly help the model behave like a sampler, which provides stability for *any* bias metric computed over model outputs (or metrics computed over distinguishable features). We do not claim this may be better than metric-specific interventions, but it does provide more freedom in mitigating bias amplification.
>
> Apart from this, we agree that it is conceptually straightforward to mitigate bias amplification by also adding more human-annotated examples. However, quantitatively characterizing the amount of reduction in bias to expect is not straightforward; we believe the fact that our bounds on model bias do seem to hold up in practice on fairly complex tasks (such as image classification and language generation) is another major contribution.
>
> > I think the intervention (how to avoid the amplification) is one key that readers would be interested in. For example, by simply overfitting the model, does it also affect the model’s generalization ability, i.e., accuracy?…. If sampling-based models are less prone to amplify bias, how does these models compare to argmax models on performance metrics? Is there a trade-off between the two metrics? This is not currently discussed, validated in the experiments.
>
> Yes we agree that comparing performance metrics between the argmax models, sampling models, and intervention models would be very interesting. Thank you very much for the suggestion! We added experiments to measure model utility two ways: 1) coherence score [Su et al 2022], which measures the similarity between prompts and corresponding model completions, and 2) mauve score [Pillutla et al 2021], which measures the difference in distributions between model-completed sentences and ground truth sentences. We updated the paper to present a full analysis of these metrics in Appendix C.2 (page 18); we now briefly discuss the main takeaways here.
>
> Comparing just the argmax model (beam search) to the sampling model (nucleus sampling), the beam search model does have a higher coherence score (0.35 vs 0.29). So in situations where coherence is valued more, such as machine translation, choosing the more repetitive beam search model presents a utility-bias tradeoff.
>
> As for our intervention of overfit beam search, we do see that the coherence is significantly decreased to 0.26 (vs 0.35). Model generalization also takes a hit: the test set perplexity jumps from 32 to 599, and the 5-gram overlap between model outputs and training data sentences jumps from 11% to 25%. However, repetition and fluency is significantly alleviated (a qualitative quick look at the sample generations in pages 22-23 confirms this). Thus, this intervention introduces a new axis to control the utility-bias tradeoff: instead of trading coherence for reduced repetition by switching from beam search to sampling, one may instead trade them off by overfitting the beam search model to different degrees. The latter may be preferred when deterministic outputs are needed but repetition needs to also be reduced.

---

> ### Author Response · Authors · 2022-11-16
> **Author response to reviewer GiPd (2/2)**
>
> > Sec 5.2 confuses me a bit about the usefulness of the proposed theoretical framework. Does it mean that the proposed bound cannot provide useful insights on real-world datasets where the calibration error can frequently be high?
>
> We would like to clarify two points here. First, while traditional calibration is frequently high on real-world datasets, consistent calibration is a separate metric and does not behave the same way. In particular, an interpolating classifier may have high traditional calibration error due to its overconfidence, but Nakkiran & Bansal [2020] show that such classifiers frequently have low consistent calibration errors. To clarify this distinction for the reader, we added an inline comment next to the consistent calibration definition and elaborate more on the differences in Appendix A (page 14).
>
> The second point is about the usefulness of the bound when consistent calibration error is high. The bound does become loose when this error is very large, as the reviewer points out. However, this still provides a valuable insight: it suggests that experimentally, bias may amplify rather quickly. Indeed, this is precisely what we see happen in Section 5.2, where the initial bias is large and amplifies another 5-7% within 5 more rounds of feedback. This experimental trend (qualitatively) matches the behavior of the upper bound, more so than the bound’s initial looseness would suggest, and so it may still be useful as a guide in these situations.
>
> > In Sec 5.3, isn’t the lower toxicity level the better? If so, why should we care about mitigating the amplification in the cost of resulting in a more toxic model?
>
> We also found this surprising, given that prior work had reported on the high model toxicity rates [Gehman et al 2020]. Instead, as the reviewer observes, the model’s toxicity level was lower than the dataset level and amplified downwards. Our claim is not that we should want to stabilize this process, but that our bounds and characterizations hold even for counterintuitive situations such as these.
>
> > Experiments are shown in the settings with 80% and 50% of new samples are model-labeled. How about the settings where <50% are model-labeled?
>
> Thank you for the suggestion! We added experiments with 20% and 5% of new samples being model-labeled, both for image classification (Figure 15) and for language generation (Figures 21 & 22).
>
> > However, I hope to see more takeaways/insights to be provided from the theoretical analysis and more experimental results outside of only validating the theoretical findings. Specifically, how one can leverage the findings to design better methods to solve real-world bias amplification problem.
>
> We believe that our detailed characterization linking consistent calibration to feedback stability will be useful as a guide for future experimental design, and our presented mitigation strategy is an initial attempt at this. We agree that more work on mitigation strategies would be very interesting and useful. In this work, our focus started with different real-world feedback settings, and we introduced just enough theoretical abstraction (e.g. consistent calibration) to derive useful upper bounds and help characterize our experiments.

---

> ### Comment · Reviewer_GiPd · 2022-11-22
> **Thank you for the response**
>
> Thank you to the authors for providing the response. I would remain my score as is given where the paper currently stands. I believe that by  better connecting the theoretical findings to actual actionable items to solve a practical problem would make the paper stronger. For example, the experiments on model toxicity is still somewhat confusing to me.

---

> > ### Author Response · Authors · 2022-11-29
> > **Updated author response**
> >
> > We thank the reviewer for taking the time to read and engage with our response. We are happy to continue the dialogue and appreciate the reviewer’s time.
> >
> > > I believe that by better connecting the theoretical findings to actual actionable items to solve a practical problem would make the paper stronger. For example, the experiments on model toxicity is still somewhat confusing to me.
> >
> > We will make sure to edit the text to better clarify what the takeaways from the model toxicity experiment are. The strength of the language experiments is in the repetition bias results (Figure 5 right), where we are able to significantly reduce model repetition while keeping generation deterministic (i.e. reduce repetition for beam search).
> >
> > We do agree with the reviewer that more experiments targeting a wider range of problems would be helpful. We view the core of this work as setting up the data feedback model and understanding when bias amplification is an issue, which includes deriving experimentally useful quantitative bounds. Though we view using this framework to derive additional new mitigation strategies as important future work, we will also add a section to the end of the paper on initial early works exploring improving consistent calibration, such as https://arxiv.org/abs/2204.03230 (which shows that differentially private training leads to consistent calibration).

---

### Decision · Program_Chairs · 2023-01-20

**Decision:**

Reject

**Justification For Why Not Higher Score:**

Reviewers raise quite a few valid concerns.

**Justification For Why Not Lower Score:**

N/A

**Metareview: Summary, Strengths And Weaknesses:**

The paper studies how dataset biases change when models are trained on some data distribution, generate more data, and are then re-trained on this generated data in addition to the original data. This problem has become very relevant with the rise of models like GPT-3 and Codex. The paper develops a theoretical framework to study this problem, and validates the framework in a set of empirical evaluations.

The reviewers value the theoretical framework the paper develops and praise the clear presentation of the work. The main concern raised is that it is unclear what the main learnings / take-aways are from the results presented in the paper. The observation that bias can be reduced by adding more examples from the ground-truth distribution to the training data and/or calibrating model predictions better appear somewhat trivial. There are also concerns about the comprehensiveness of the theoretical analysis.

While the present manuscript is not ready for acceptance due to these concerns, the AC encourages the authors to revise their manuscript and resubmit it to a future conference. The reviewers provided a lot of useful feedback that ought to help the authors in strengthening their manuscript.